# Intervallic intonation: Applying the Implication-Realization model of musical melody to speech intonation and prosody

**Alfred W. Cramer** *

Department of Music, Pomona College, Claremont, California, United States of America

* acramer@pomona.edu

## Abstract

This methodological study presents the Implication-Realization (IR) model as a framework for the analysis of linguistic prosody and examines its application to English-language examples of speech. Originally developed by Eugene Narmour for music analysis, IR's cognitively-based approach views melodies as hierarchical structures formed through processes of implication and closure. It parses melodies by comparing successive pitch intervals while also considering duration and potentially other parameters. With computational assistance from a newly developed set of Praat scripts (IRProsodyParser), the study applies an adapted version of IR's symbology to several Modern American English examples. In this adaptation, comparisons of successive pitch intervals form the basis for a categorical classification of interval sizes. IR-generated parsings show broad correspondence with those produced within the autosegmental-metrical (AM) framework, with AM boundary tones, phrase accents, and pitch accents manifested at progressively deeper levels in the IR hierarchy. These findings support the view that pitch intervals are central in perceiving speech intonation and that intonational features arise as the result of a complex interaction of pitch, duration, and other cues. Moreover, while AM and similar approaches often frame intonational features in terms of aural prominences within the melodic stream, IR encourages viewing them in terms of their positions within a melodic hierarchy.

## Introduction

A brief early statement of Eugene Narmour's Implication-Realization model argued that the model could be applied to speech intonation as well as musical melody [1]. Speech intonation and prosody have often been compared to musical melody, with the patterns of spoken intonation cast as "tunes" to which words are fitted [2,3]. This analogy has at times faced skepticism, for example on the grounds that pitch ($f_0$) invokes fundamentally different structures and functions in music and speech, or that

**Data availability statement:** The data analyzed are available are available in the paper and its Supporting information files. Specifically, they are available in Brugos, A. et al (2006). Transcribing Prosodic Structure of Spoken Utterances with ToBI. MIT Open Courseware. https://ocw.mit.edu/courses/6-911-transcribing-prosodic-structure-of-spoken-utterances-with-tobi-january-iap-2006/pages/lecture-notes/. This source is cited at appropriate points in the article. The findings of the article do not depend specifically on such data, which are used for illustrative purposes. In creating many of the illustrations, I have used computational scripts created by myself to be used in the linguistic software Praat. The scripts are available at acramer.sites.pomona.edu/IRProsodyParser.

**Funding:** The work was supported by a sabbatical subvention to AWC and support for a student research assistant, both provided by Pomona College (https://www.pomona.edu). No grant numbers were associated with this funding. The funder did not play any role in the study design, data collection and analysis, decision to publish, or preparation of the manuscript.

**Competing interests:** The author has declared that no competing interests exist.

the durational structuring of linguistic units such as words or syllables is not isochronous like the structuring of musical tones [4]. Indeed, approaches to linguistic prosody in recent decades have tended to set aside musical comparisons.

This study, however, demonstrates that Narmour's Implication-Realization model of musical melody (IR) [5–7] can be effectively adapted for speech parsing so as to address key issues such as hierarchical organization within intonational contours, boundaries and breaks between their elements, completeness in parsing, and conveyance of information structure. The article begins with overviews of the well-established autosegmental-metrical approach to speech intonation (AM) and of the IR model. It next introduces an adaptation of IR that facilitates its application to intonation, along with a set of computational scripts, IRProsodyParser, which assist in that application. Three case studies follow, with IR-based parsings of spoken utterances examined against the background of AM. The article concludes by considering IR's potential as an explanatory framework for speech intonation.

Together with prosody (the patterning of spoken stress and duration), intonation involves a complex interplay of pitch ($f_0$), duration, intensity (loudness or stress), and other types of features that interact to create prominence relations within and between phrases and to signal phrase boundaries [8]. Mid-twentieth-century accounts treated intonation as a suprasegmental kind of "speech melody" [9,10] that floated above phrases. It was thought that "intonation does not change the meaning of lexical items, but constitutes part of the meaning of the whole utterance" [11]. From the 1960s to the 1980s, however, linguists came to recognize that the intonational meaning of an utterance lay in the alignment of the most informative lexical items to specific features of intonational melodies [12–14]. This shift recast the pitch contours of linguistic intonation as a series of inflections aligned to words. Since the 1980s it has been established that intonational patterns are phonologically structured and convey signals about discourse and information structure, aiding listeners in grasping the significance of words and phrases within a discourse [15,16]. Intonational linguists often consider spoken tones in terms of how they signal whether a phrase is an assertion or a question, whether it is concluded or will continue, and whether a word conveys new or established information.

### Intonational features as prominences in the melodic stream: The autosegmental-metrical (AM) and other linguistic approaches

The most widely-known approach to intonation, known as the autosegmental-metrical approach (AM), describes intonational pitch primarily in terms of two broadly defined pitch levels, high pitch (symbolized **H**) and low pitch (**L**), while, more importantly, accounting for the functions of the pitches [8,14,16–18]. Fig 1, an analysis of a recorded utterance used in training materials for the AM-based Tones and Break Indices annotation system (ToBI), labels the most salient tones of intonational phrases according to function [19,20]. Phrase accents [**-**] and boundary tones [**%**] convey information about a phrase's structural ending and its connection to surrounding phrases; in English they often occur together, as at the end of Fig 1 where the falling pitch symbolized **L-L%** indicates that a declarative sentence has concluded.

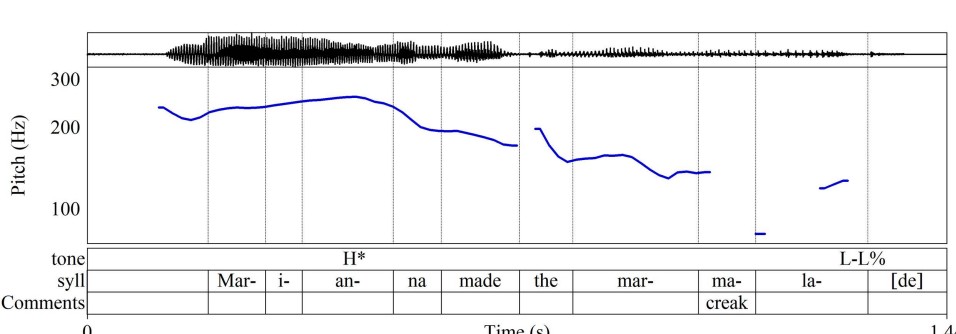

**Fig 1. AM annotation of an utterance of "Marianna made the marmalade." Annotation after [19,20].**

Pitch accents [*] signal the informational significance of the words to which they are attached: a high pitch accent **H\*** on a word's stressed syllable typically conveys that a word conveys new information; a low pitch accent **L\*** marks information already understood in a discourse. In this production of the phrase, the **H\*** on *Marianna* indicates that the speaker believes the listener does not already know who made the marmalade; perhaps the utterance is a response to the question "Who made the marmalade?" (Throughout this article, stressed syllables are underlined.)

The **H\*** on *Marianna* and the L-L% ending in Fig 1 may adequately represent the overall effect of the phrase's intonation, but other features in the contour warrant attention as well. In particular, to omit labels for *made* and for *mar-* in *marmalade* is to disregard pitch movements—such as the local peak on *mar-* and the rise from *-ma-* to *-lade*—that could be intonationally relevant. It is common to regard such pitch movements as interpolations between salient tones, explained as habitual phonetic patterns that are phonologically insignificant [21–23]. Yet such interpolations may still play a role in the contour's construction and thus warrant discussion. One might argue for labeling *mar-* with **L\***, since it anchors a word whose meaning is already established in the discourse and it is substantially lower in pitch than the **H\*** of *Marianna*. However, such a judgement should be informed by other measures of contoural significance, not only pitch level. A fuller account of the relationship between pitch and other parameters in the shaping of the pitch contour is required.

In music, pitch is at the forefront of awareness. It is usually steady, and it is identifiable or describable (for example, by names or scale degrees). By contrast, accounting for the perception of pitch in speech is challenging [2]. Speakers and listeners often have only a vague awareness of it. The extent to which the perception of spoken pitch invokes precise pitch targets, approximate pitch areas, or features such as peaks, valleys, rises, and falls remains contested, as does the degree to which spoken pitches are conceptualized in relation to each other versus to a background frame of reference (that is, syntagmatically versus paradigmatically).

Intonation has both syntagmatic and paradigmatic aspects, but the paradigmatic is undertheorized, especially regarding the scaling or gradient variation of relative pitch heights [8,24]. AM's basic levels **H** and **L** are defined in relation to an external frame of reference such as the speaker's vocal range. It has been argued that AM's emphasis on a paradigmatic or level-based approach reflects the lack of a suitable syntagmatic approach at the time the theory was being formed [21,25]. Even so, AM pitch is often syntagmatic in practice—for example, in the conception of downstep, in the estimation of a speaker's vocal range from the pitches used in an utterance, or in the assignment of **H** and **L** labels to local peaks and valleys. Other transcription systems, such as INTSINT [26,27], combine syntagmatic and paradigmatic labeling with greater transparency and variety, in addition to labeling more features of the pitch contour. The AM+/RaP reformulation of AM/ToBI [21,25,28] treats pitch accents as sequences of multiple tones representing successions of pitch targets. Its labels mark tones by interval category rather than pitch level: whereas AM's H denotes a high tone, AM+ uses H to

indicate a rise from the preceding one. Further, because it attributes one (or more) tone per syllable and places pitch accents on all stressed syllables, AM+/RaP would mark pitch accents on *made* and *marmalade* in the example discussed above.

Still, all of these approaches treat the intonational contour as a single-level signal to be parsed through a search for features such as pitch accents and boundary tones. Typically the features have to do with prominence. An overemphasis on pitch may lead to an equation between high pitch level and aural prominence [21], while a broader view recognizes that AM's **H** and **L** are mental categories that may be instantiated not only in pitch levels but also in properties such as duration or loudness [8]. Regardless, the hierarchy that emerges is one of function: pitch accents are mapped to words, boundary tones to intermediate phrases, phrase accents to intonational phrases.

### Melody as hierarchically organized stream: the Implication-Realization model (IR)

By contrast, the model presented here, IR, parses the melodic contour into an event hierarchy without considering linguistic function. Its groupings of tones are based on comparisons of pitch, duration, and other features within the contour. The initial and final tones of these groups emerge as events on the next hierarchical level, which in turn is again organized into groupings. This is the kind of hierarchy encountered in music. We will see that the accents and tones of AM broadly correspond to certain types of events in the IR hierarchy, but with tones that emerge at deeper levels not chosen for their aural prominence—particularly not their prominence in pitch—nor for their significance in a linguistic hierarchy.

Like AM+, IR considers the pitch of every tone, and it does so entirely in relation to the other tones—it is natively syntagmatic. It treats the constantly fluctuating actual pitch of speech as produced and perceived through a series of discrete tones with target pitches. (For support of this approach, see [29,30]). Rather than labeling pitches, it labels intervals, using a categorization system that is couched in terms of distinctiveness and sensitive to context. This approach has a precision appropriate to intonation and not modeled on musical nomenclature.

Just as importantly, IR's concepts of implication and closure and its hierarchical structure may well illuminate the mechanisms by which tones of speech convey spoken phrase and information structure through the prosodic alignment of word stress and pitch accent. In AM, the functions of different tones are explained principally in terms of assigned meaning—of being learned with the language. IR, however, explains them structurally. Boundary tones are implicative tones at shallow hierarchical levels that leave groupings incomplete while signaling expectations about how the beginnings of subsequent phrases may complete them. Pitch accents are deep-level tones—just one or two per phrase—whose signaling of informational significance can be explained in terms of their implicative connections to the deep-level tones of neighboring phrases.

IR is used here in a form modified orthographically from Narmour's versions of the model so as to emphasize the model's capability of interval categorization. Crucially, the interval categories are not those usually associated with music. IR is fundamentally concerned with basic perception rather than with musical qualities such as chroma or learned formal categories such as such as major 3rds or perfect 5ths. Thus, IR categorizes intervals with a view to whether each succession of two tones is distinctive in size or direction compared to the immediately preceding succession. The thresholds for distinctiveness are systematically context-sensitive. The interval categories, in conjunction with the tones' durations (and potentially other factors) influence whether a grouping of tones is complete or more is to be expected.

**Background of IR.** Formalized by Narmour [5,7,31,32], the IR model is rooted in Meyer's ideas about Gestalt perception, musical schemata, and the aesthetics of expectation, surprise, and satisfaction [33,34]. Meyer introduced implication as a principle involved in the building of musical continuity across time: "An implicative relationship is one in which an event … is patterned in such a way that reasonable inferences can be made both about its connections with preceding events and about how the event itself might be continued and perhaps reach closure and stability" [34]. One may think of an *implication* as an expectation regarding the tones likely to continue a melodic grouping beyond the current tone. *Closure*, the opposite of expectancy, refers to the sense that the current tone has completed a melodic grouping, suppressing further expectation and bringing about stability.

Music analysts have long focused on factors such as harmonic consonance and dissonance or metric strength and weakness when examining musical stability. A major contribution of IR is its proposal to add *reversal of direction* and decrease in *rate of pitch motion* to the list of factors contributing to such stability. The cognitive processing of these phenomena is considered bottom-up—that is, stimulus-driven, automatic, and always active even when operating alongside learned or conscious (top-down) processes such as the evaluation of musical harmony. IR also posits that differences in successive tones' durations, loudness, and other perceptible features exert bottom-up implication or closure. If a tone's closural features outweigh its implicative features, a grouping of tones is closed. These basic principles of implication and closure form the foundation of IR's hierarchical approach to melodic structure, where initiating and closing tones of melodic groupings are identified at progressively deeper analytical levels. At the same time, as a comprehensive theory of melody, IR accounts for top-down (knowledge-driven) patterns of perception and formalizes the ways in which these patterns may influence or modify bottom-up parsings.

Numerous empirical studies have investigated IR's basic hypotheses about bottom-up pitch expectation in music [35]. In general, these studies have confirmed IR's claim that a small interval creates the expectation of another in the same direction while a large interval implies a small interval in the opposite direction [36–41], although some have advocated a simplification of the model [42,43] (but see [40,41]), some have argued that IR's basic expectations are learned rather than innate [43,44], and some have claimed that the expectation of a reversal after a large interval is best explained by constraints on pitch range—especially vocal pitch range—rather than the cognitive processing of pitch patterns [45,46]. IR helped lay the groundwork for a flourishing subfield of theorization about heightened anticipation in music [47,48].

Yet IR was initially developed as a contribution to the study of musical event hierarchies [49,50]. Early reviewers regarded it as a comprehensive system of musical analysis distinguished by its particular account of melodic connection and hierarchy [31,37,51–54]. They valued its approach to pitch, even while noting the difficulty of isolating the bottom-up perception of pitch intervals from the many other aspects of musical syntax [51,55]. Few empirical studies have explored IR's hierarchical aspect—a rare exception is [56], discussed in [57]—but it is acknowledged that simply investigating expectation alone does not address IR in its full complexity [43,44].

This study takes IR to be fundamentally a model of hierarchical melodic structure. It is most engaged with the premise that similarities and contrasts between intervals add to or subtract from closure and that closed tones are also opening tones of the succeeding grouping and emerge on the next hierarchical level [5]. Through three analytical case studies, it will offer a proof of concept of IR's applicability to intonation and prosody.

## Method: Parsing prosody using IR

In IR, each grouping starts with zero implication. Implication is added by the second tone of a grouping, and added, subtracted, or left the same by each note thereafter until total implication either returns to zero or is offset by a secondary factor such as an increase in duration. When closure returns to zero, the grouping is closed. A tone increases implication if the interval it forms with the preceding tone is distinctly larger than the interval formed by the the two preceding tones; it decreases implication (or adds closure) if it forms a tone that is distinctly smaller. Each tone is considered in terms of the interval it forms with the preceding tone (if any) and the comparison of this interval with the interval before it. Separately, its duration is compared to the duration of the preceding tone. If this evaluation ascertains that the grouping is closed, its final tone advances to the next deeper structural level. This process is applied iteratively to all tones across every level.

IR analysis, then, involves quantitative, comparative consideration of each tone's pitch and duration. Because of music's discrete pitches, interval sizes, and durations, a human analyst can fairly easily perform the calculations to decide whether a difference in interval size or tone duration is distinctive. Thus it is easily possible to produce an IR analysis of music without computational assistance. (See S1 Text in Supporting information for a demonstration of musical IR analysis.) However, in speech, the comparisons of intervals and durations are not arithmetically simple, so it is useful to have computational assistance. The analyses here have been created using IRProsodyParser [58], a semiautomated

computational algorithm scripted in Praat by the author [59]. IRProsodyParser serves two purposes. First, it facilitates the close aural and visual inspection needed for the user to find the beginning, end, and target pitch of each tone. Second, using that information about the tones as input, it performs the iterative calculations that lead to a completed multi-level diagram of the utterance's IR hierarchy.

### Analyzing with IRProsodyParser: A demonstration

Before examining the principles of IR in detail, it will be useful to illustrate the creation of an analysis through IR. This application of IR to speech assigns at least one tone to each syllable and considers every tone in its representation of the prosodic hierarchy. The first task in analysis is to find the onset and end of each tone and the pitch of each tone. IRProsodyParser helps the user accomplish this with aid from Praat's tools such as spectrographs and easy replay of segments. Fig 2(a) shows a screenshot of this stage of the segmentation.

Because the purpose of finding the onset of each tone is to represent its duration in a melodic sense, the task should not be treated as a matter of orthographic or morphological syllable-division. Rather, the onsets should mark each syllable's P-center [60,61]—a robust phenomenon but one whose acoustic and articulatory cues remain elusive [62,63]. Onsets are measured from the point of articulative release that initiates the sounding of a syllable's nuclear vowel to the onset of the next syllable's nuclear vowel or a pause. In Fig 1, such releases include the instants when lip contact ends in the *m* sounds of *Marianna* and *marmalade*, when tongue contact ends following *n* in the final syllable of *Marianna* and the

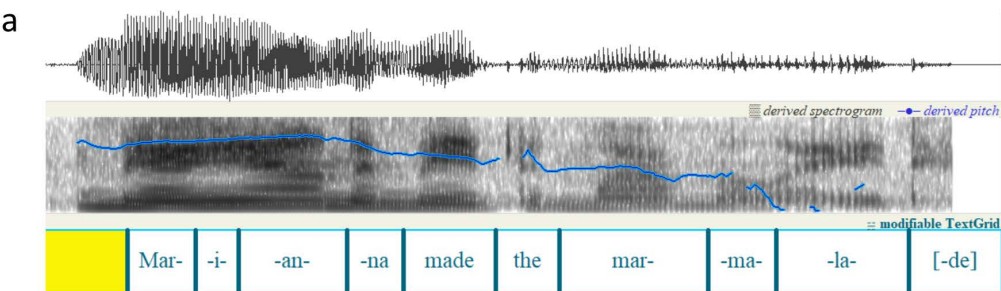

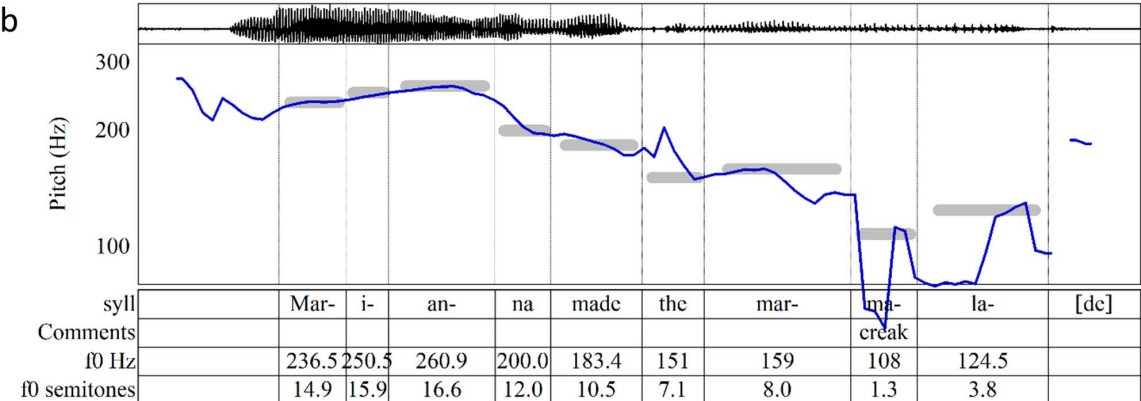

**Fig 2. Demonstration: syllabification and pitch assignments of "Marianna made the marmalade" using IRProsodyParser.** (a) The spoken signal is segmented into syllables by the user; a screenshot from Praat shows the waveform, spectrogram, and pitch trace, with syllable boundaries marked in a TextGrid. (b) Pitch values are identified by the user with computational assistance and entered either in Hz or semitones relative to 100 Hz; the TextGrid displays pitches in both formats, which are expressed as grey bands on the graph.

*d-th* of *made the*, and when the airflow of *i* is restored to launch the of *-a-* of *Marianna*. In the final syllable *-lade*, the audible release of tongue contact of *d* marks the tone's end and the beginning of a break in speech, even though the start of the *d* obstructs airflow and stops the sound earlier. A tone's duration is the time from its onset until the onset of the following tone or silence.

An accurate estimate of each tone's target pitch is crucial. A trained musician can identify the pitches aurally, or one may turn to software such as Prosogram [64] to estimate the pitches of tones. For this proof-of-concept study, the target pitch of each tone was identified through careful, repeated listening aided by several functions in Praat and IRProsodyParser, and entered into the Praat textgrid, as seen in Fig 2(b). The estimates used here were initially approximated by IRProsodyParser by dividing the syllable nucleus into short time slices and averaging the pitches of the slices, with each weighted according to its intensity. These approximations were then refined through careful listening. (IRProsodyParser lets the user edit a tone's pitch in either frequency [Hz] or semitones).

To verify estimates of the pitch targets in these analyses, recordings were resynthesized in Praat to give each syllable a single sustained pitch. Such stylized speech [see 48,49] may be considered a form of analytical reduction and were evaluated according to the standard set forth in the music-theoretical work of Lerdahl and Jackendoff: a reduction "should sound like a natural simplification at the previous level" [65]. Visual estimation of target pitch from the $f_0$ contour is unreliable. Some target pitches lie in the middle of the graphed pitch contour, some are asymptotic to the contour, and some are local maxima or minima. (This is partly because consonants often pull the pitch up or down [66] and sometimes because vocal creak eludes the pitch-tracking software.) Even so, it is useful to check pitch estimates by verifying that there is some visible correspondence between each target and the pitch contour as graphed in Fig 2(b).

The frequencies given in Hz and semitones and graphed as grey bands in Fig 2(b) are fed into the algorithm used by IRProsodyParser for generation of hierarchical diagrams.

The diagrams employ two sets of symbols. One set shows implication and closure. Arrows represent implications (increased expectancies) while arrowtails represent closure (that is, negative expectancy, which may be visualized as the capture of expectancy) brought about by pitch motion. Commas and inverted commas denote increases and decreases in closure brought about by other features. These changes in expectancy are summed to determine whether a given tone closes a grouping. Beginnings and endings of groupings are marked by square downward brackets. The second set of symbols indicates the relative directions and sizes of intervals.

Fig 3(a) illustrates the parsing of a grouping. The first tone is marked with an opening bracket. Turning to the second tone, the algorithm determines that the first two tones form a large descending interval, labeled **L** for *large*, with italics to indicate the descent in pitch. It then marks the implication added by the tone. As with any non-zero initial interval, **L** receives two arrows to indicate implications of both direction and size. The following interval (*-ma-lade*) is labeled **R–**: **R** (for *reversal*) because it changes direction (moving up) and **–** because it is distinctly smaller than the previous interval. Both features are closural, so the final tone receives two arrowtails. The tails capture the inherited expectancy of the previous tone's two arrows, reducing the net implication on *-lade* to zero.

The grouping may not yet be called closed, since features producing secondary forms of closure (denoted by commas) and counterclosure (denoted by inverted commas) must still be considered—specifically, effects from relative duration and direction. These secondary effects apply only to the tones on which they occur: unlike arrows and tails, the commas and inverted commas that denote them do not cumulate throughout a grouping. In Fig 3(a), the syllable *-ma-* is distinctly shorter than the preceding syllable, making it less likely to be closural. This counterclosural effect is denoted by an inverted comma ['] above the arrows. At the same time, the descent of the **L** adds a comma, because large descents have closural effect. Whether the grouping ends here is determined by summing up the algorithm sums up the arrows, tails, inverted commas, and commas that apply to the tone, as shown in Table 1. With total expectancy above zero, the grouping stays open.

The next tone in Fig 3(a), on the syllable *-lade*, is very much longer than *-ma-*. Its duration suppresses implication, adding three commas of closure. The calculation of expectancy on this tone, seen in Table 2, reflects all implicative effects

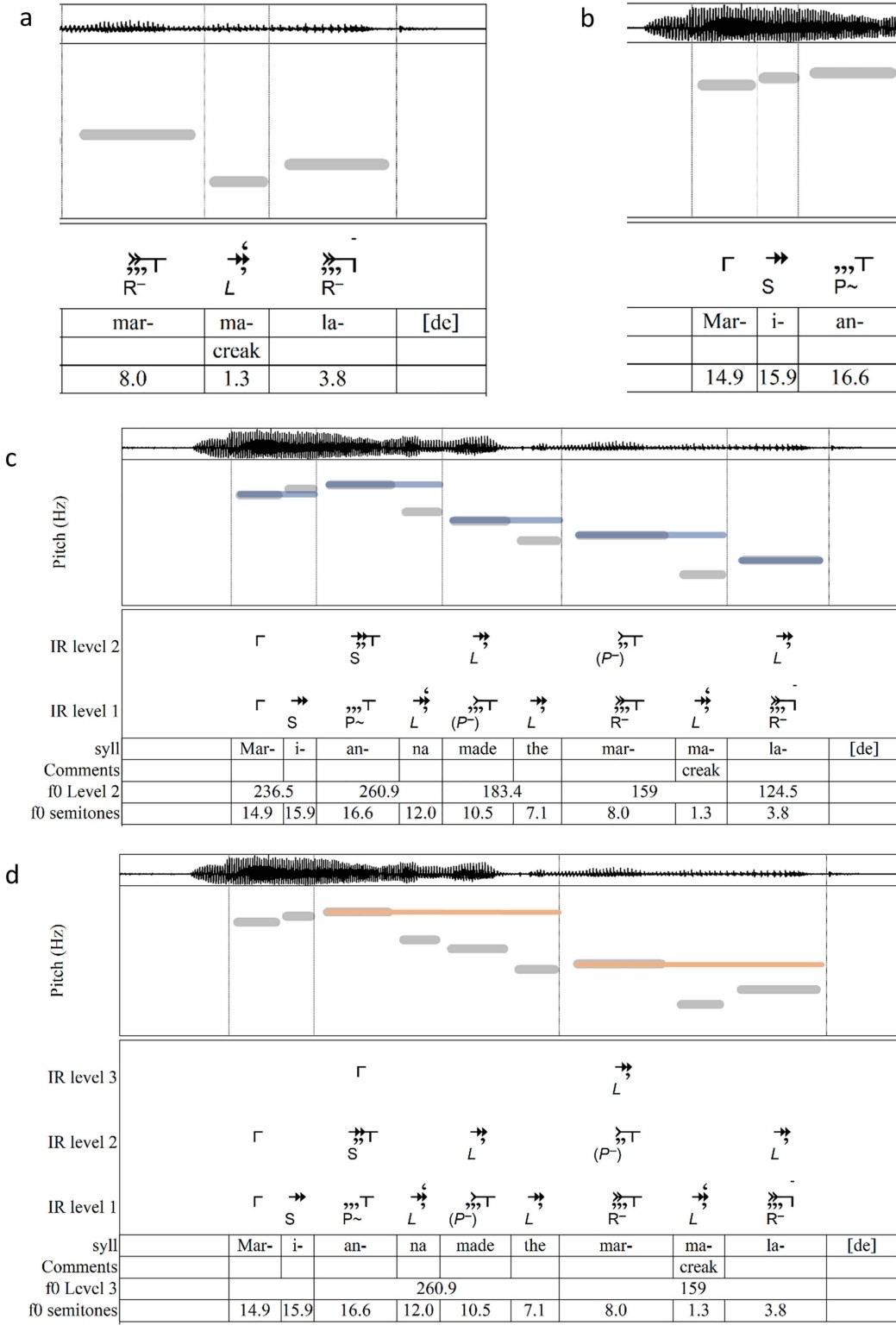

**Fig 3. Demonstration: parsing of "Marianna made the marmalade" using IRProsodyParser.** Panels show automatic parsing by IRProsodyParser according to the Implication–Realization algorithm: (a) close-up view of an L R– grouping; (b) close-up view of an S P⁻ grouping; (c) parsing of Levels 1 and 2, where end tones of Level 1 groupings give rise to emergent Level 2 tones shown as blue bands; and (d) parsing of Levels 1 through 3, where end tones of Level 2 groupings give rise to emergent Level 3 tones shown as orange bands.

(arrows and tails) since the start of the grouping, but only the tone's own secondary effects. With negative total expectancy, the tone closes the grouping and is marked with a closing bracket.

Fig 3(b) illustrates the parsing of a differently structured grouping. The first two tones (*Mar-i-*) form a small ascending interval **S**, which, like any non-zero initial interval, contributes two arrows of implication. The next interval (*-i-an-*), having the same direction and approximately the same size, is labeled **P˜**: **P** (for *process*) for continued direction and **˜** for continued size. Since **P˜** follows the established trajectory, it adds neither implication nor closure and receives no arrows or tails. In terms of duration, *-i-* is shorter than *mar-*, but not by enough to meet the threshold of distinctiveness, so it receives no inverted comma. The tone of *-an-*, however, is so long that it receives three commas of closure. These three commas offset the two inherited arrows, so the grouping is closed and marked with a closing bracket. This tone also initiates the next grouping, and thus carries an opening bracket. (The symbol seen above *-an-* combines those two brackets.)

The labeled groupings in Fig 3(a) and (b) are seen in context in Fig 3(c), which illustrates the formation of an emergent level (Level 2, with tones graphed as blue bands) from the surface level (Level 1). Each start or end of a grouping in Level 1 appears on Level 2 as a tone sharing the corresponding tone's Level 1 pitch and onset time. Each emergent tone ends at the onset of the next emergent tone or the next silence.

The tones of any emergent level are parsed into groupings in the same way as those at the surface level. On Level 2, groupings end on *Marianna* and *marmalade*.

Fig 3(d) depicts the same process extending to Level 3. Here, only two tones remain, *Marianna* and *marmalade*. (The initial tone, *Marianna*, does not emerge in Level 3 because the resulting grouping, **S** connecting *Mar-* to *-an-*, would merely replicate a structure already present on Level 2.)

## The adapted IR symbology

In the foregoing analysis, each tone is parsed with respect to the relative size and direction of the interval it creates, allowing for an assessment of its implicative and closural properties. This study adheres to the substance of IR as presented by Narmour, but it introduces orthographic changes that foreground each tone's implicative status and more clearly represent interruptions, suppressions of implication, and the like.

One orthographic change has to do with the distinction between expectation, which refers to the anticipation of a specific or narrowly specified pitch, and expectancy, which refers to the sense that more is to come. In Narmour's

**Table 1. Calculation of expectancy for tone 2 in Fig 3(a).**

| Type of change in expectancy | Quantity | Effect on expectancy |
|---|---|---|
| Arrows thus far in grouping | 2 | +2 |
| Tails thus far in grouping | 0 | 0 |
| Inverted commas on this tone | 1 | +1 |
| Commas on this tone | 1 | −1 |
| Total expectancy | | +2 |

**Table 2. Calculation of expectancy for tone 3 in Fig 3(a).**

| Type of change in expectancy | Quantity | Effect on expectancy |
|---|---|---|
| Arrows thus far in grouping | 2 | +2 |
| Tails thus far in grouping | 2 | −2 |
| Inverted commas on this tone | 0 | 0 |
| Commas on this tone | 3 | −3 |
| Total expectancy | | −3 |

presentation, arrows indicate that an interval carries the expectation that the following interval will be of a particular direction and size range. A tail indicates that the implication is realized; no tail is given if the implication is denied. In the present study, however, arrows represent increases in expectancy and tails represent increases in closure. This makes the calculation of closure and expectancy more visible. Moreover, tracking changes in expectancy through the use of arrows and tails is particularly valuable in the analysis of intonation, where expectancy or closure attached to a specific word may signal something about its discursive or informational significance.

**Categorizing and labeling intervals and their expectancies.** With this knowledge, we are in a position to examine in more detail the IR interval labels that were introduced above. Fig 4 provides a comprehensive list of the IR interval

| Symbol | Meaning | Associated implication or closure | Implied subsequent interval |
|---|---|---|---|
| **Initiations** | | | |
| S | small initial interval | ↠ | P~ or P⁰ |
| L | large initial interval | ↠ | R |
| 0 | Initial interval of magnitude 0 | → | D or P~ |
| **Continuations** | | | |
| **Comparative direction** | | | |
| P | Process (continuation of previous interval's direction) | — | |
| R | Reversal of previous interval's direction | ⊢ | |
| **Comparative size** | | | |
| + | Interval contrastively larger than previous | → | |
| - | Interval contrastively smaller than previous | ⊢ | |
| ~ | Interval non-contrastively different from previous | — | |
| 0 | Interval precisely same size as previous~ | — | |
| **Other** | | | |
| *L, P⁺, R⁺* | Italics denote descending interval (large or increasing) | , | |
| *S, P~, R~* | Italics denote descending interval (small or non-distinctive) | | |
| ( ) | Retrospective continuation interval (directional continuation not implied by previous interval) | | |
| P̲ | Continuing interval following magnitude-zero interval | — | |
| R̲ | Zero-magnitude continuation after non-zero interval | ⊢ | |
| D | Second consecutive repetition of pitch, synonymous with P̲⁰ | — | |
| | Unit of secondary closure contributed by increased duration, downward motion, or other non-intervallic means. | , | |
| | Unit of secondary counterclosure contributed by decreased duration or other non-intervallic means. | ' | |

**Fig 4. Implication-Realization interval labels.**

labels; Fig 5 gives expectancy and grouping symbols such as arrows, tails, commas, and brackets; and Fig 6 shows the correlations between interval labels and expectancy symbols. For now, we are interested in how to label a continuation interval (the interval between the second and third tones of a three-tone combination) in comparison to the initiating interval (the interval between the first two tones). The left-side column of Fig 6(a) shows the directional aspect. **P** is the symbol applied to a continuation that moves in the same direction as the initiation. It adds neither expectancy nor closure, so a line or dash symbolizing implicative neutrality is placed over the third tone. A reversal in interval direction is symbolized **R** and results in decreased expectancy, which is represented by a tail. The top header row of Fig 6(a) shows the interval-size aspect. A distinct increase in size is symbolized **+** and brings about increased expectancy, denoted by an arrow over the third tone. A distinct decrease has **–** in the label and decreased expectancy (increased closure), represented by a tail. A continuation interval with only a small size difference from the preceding interval is implicatively neutral, receives no arrow or tail, and has ˜ as part of its label if the difference is non-distinctive—that is, small enough to be regarded as insignificant—or 0 if the size difference is unnoticeable.

The body of Fig 6(a) shows the direction and size symbols combined: **P+**, **R˜**, and so on. These symbols represent a comprehensive set of mental categories for intervals in that they show whether or not pairs of tones, when compared to the preceding pairs, are recognized as similar or distinct from each other in direction and size.

Categorical thresholds separating distinct from non-distinct differences depend on context and judgment and likely vary among speakers and listeners. In modeling them, IRProsodyParser permits adjustment of these thresholds, but requires them to be kept constant over the course of an analysis. Most spoken intervals are smaller than four semitones, clustering near zero, while music has more large intervals, peaking near two semitones [23,67]. The thresholds between ˜ and **+** or **–** are thus likely smaller in speech than in music. 't Hart found that when listeners compared pairs of spoken intervals

| Symbol | Name | Meaning | Effect on parsing |
|---|---|---|---|
| | | **Implication and closure** | |
| ⌐ | Opening bracket | Initiation of melodic grouping | Sets total implication equal to 0. |
| ⌐ | Closing bracket | Conclusion of melodic grouping | Applied when total implication ≤ 0. |
| ⊤ | Combined opening and closing bracket | Used on a tone that is both an initiation and a conclusion | |
| → | Arrowhead | Implicative interval feature. Increase in expectancy. | +1 added to total implication for every note until end of grouping. |
| ⊱ | Tail | Closural interval feature. Decrease in expectancy. | -1 added to total implication for every note until end of grouping. |
| — | Neutral tone | No new implication or closure | No effect. |
| , | Secondary closure | Increased duration or another secondary feature is a factor contributing closure. | -1 added to total implication for this tone only. |
| ' | Secondary counterclosure | Decreased duration or another secondary feature is a factor working against closure. | +1 added to total implication for this tone only. |
| ] | Suppression of implication | Interrupts; concludes grouping without closure. | Overrides total implication score and assigns { to next tone. |
| ⌐ | Releasable suppression of implication | Interrupts; grouping may resume after pause. | No effect. Overrides total implication score if ≤ 0. |

**Fig 5. Implication-Realization grouping symbols.**

**Fig 6. Interval labels and expectancies.** (a) Continuing intervals are labeled by comparative size and direction, with implicative and closural effects symbolized by arrows and tails. (b) Initial intervals of groupings are categorically labeled and arrows of implication assigned according to size.

presented in consecutive stimuli, differences under about 0.3 semitones fell below the "point of subjective equality" while also concluding that "only differences of more than 3 semitones play a part in communicative situations" [68]. The first value may guide the choice of the boundary between **P⁰** and **P˜** or **R⁰** and **R˜** (although the current version of IRProsodyParser cannot set this value independently from the boundary between initial **0** and **S**). The second value gives a range for the boundary between **˜** and **+** or **−**. As for durations, in speech they are irregular and variable and their measurement likely more sensitive than in music. This study uses the thresholds shown in Table 3.

In addition to labels for continuation intervals, the initial intervals of groups must be labeled. The present adaptation labels the initial tones of groupings as illustrated in Fig 6(b): **L** for large, **S** for small, and **0** if the second tone repeats the pitch of the first. (These labels are not in Narmour's original, where labels identify three-tone structures by comparing the second label to the first, but they are implicit in Narmour's important distinction between the expectations aroused by large vs. small initial intervals.) Each of these initiations carries expectancy denoted by arrows. **L** and **S** receive two arrows, while **0** receives just one.

**Table 3. Provisional threshold values of IR's intervallic categories.**

| Threshold | Definition in music [5,6] | Definition in speech posited in the present study |
|---|---|---|
| 0 | $i_i = 0$ semitones | $i_i < 0.2$ semitones |
| S | $i_i < 5$ st | $i_i \leq 2.5$ st |
| L | $i_i \geq 5$ st | $i_i > 2.5$ st |
| P− | $i_1 - i_2 \geq 4$ st | $i_1 - i_2 \geq 2$ st |
| P+ | $i_2 - i_1 \geq 4$ st | $i_2 - i_1 \geq 2$ st |
| P⁻ | $|i_2 - i_1| < 4$ st | $|i_1 - i_2| < 2$ st |
| P⁰ | $i_1 = i_2$ [a] | $|i_1 - i_2| < 0.2$ st |
| R− | $i_1 - i_2 \geq 3$ st | $i_1 - i_2 \geq 1.5$ st |
| R+ | $i_2 - i_1 \geq 3$ st | $i_2 - i_1 \geq 1.5$ st |
| R⁻ | $|i_2 - i_1| < 3$ st | $|i_2 - i_1| < 1.5$ st |
| R⁰ | $i_1 = i_2$ [a] | $|i_2 - i_1| < 0.2$ st |

$i_i$ = size of initial interval of a grouping

$i_1$ = size of first interval of a pair

$i_2$ = size of second interval of a pair.

[a]P⁰ is a special case of P⁻ and R⁰ is a special case of R⁻.

Three additional symbols shown in Fig 4 round out the system. (1) Following Narmour [7], descending intervals are italicized. (2) Retrospective intervals. These are intervals in which expectations are violated (as when **L** is followed by **P**, or **S** by **R)**, forcing retrospective re-evaluation of the initial expectation. The retrospective aspect has no effect on closure. (3) If three consecutive tones have the same pitch, the second interval is labeled D, for *duplication*. Underlinings denote cases of two consecutive tones with the same pitch.

**Calculating closure: Arrows, tails, and secondary features.** Although arrows and tails can arise from both intervallic and registral implications, they represent expectancy generically and are summed together. An arrow represents a quantum of expectancy, a tail a quantum of closure (negative expectancy). They are tallied starting from the initial tone of a segment. When a melodic grouping's total implication descends to 0 or less (i.e., when the tally of tails balances the tally of arrows), all expectancy is satisfied, the structure is closed, and the next interval is evaluated as an initiation. Otherwise, it is evaluated as a continuation.

However, IR parsings are usually nonsensical if features such as duration and downward motion are not considered. These features, here termed *secondary*, have their own closural and counterclosural effects. In this adaptation of IR, any closure contributed by a secondary feature is symbolized by a comma [,] equal to the closure of one tail, while any instability is symbolized by an inverted comma ['] and is equal to that of one arrow. IR regards durational change as a secondary feature. A tone distinctly longer than the preceding tone receives a comma of closure; if it is a great deal longer, it may receive two or even three commas. Intuitively, greater increases in duration have greater closural effect; however, the premise of this study is that features are evaluated in terms of distinctiveness, not scaled quantity. The solution in this study is to classify durational increases distinctive [,], highly distinctive [,,], and exceedingly distinctive [,,,], according to the formula

$$number\ of\ commas = \left\lfloor \log_c \frac{d_c}{d_p} \right\rfloor$$

(1)

where $d_c$ is the duration of the current tone, $d_p$ is the duration of the preceding tone, and $c$ is a constant equal to the durational increase that produces a single comma of closure. As a guideline for quantifying secondary closure, Narmour

estimates that, in music, a tone 1.5 times as long as the preceding tone normally closes a grouping of **L P˜** [5,6]—that is, it is as closural as if it had two tails. To achieve this, *c* may be set to 1.225 for music; for speech, it is set to 1.2 in this study. Conversely, a tone shorter than the preceding tone is counterclosural and receives inverted commas according to the formula

$$number\ of\ invertible\ commas = \left\lfloor \log_r \frac{d_p}{d_c} \right\rfloor \tag{2}$$

where *r* is a constant generally set to 0.8 for speech in this study.

In addition, descending **L** intervals receive a comma of secondary closure, as do descending **+** intervals [6,69].

Secondary closure and expectancy apply to single tones only; their effects do not carry over to later notes as they do with interval-based expectancy. Thus, commas and inverted commas are not tallied over the span of a whole segment like arrows and tails but contribute only to the tally of a single tone. The closure on a single tone, therefore, may be described thus:

$$net\ implication = (a - t) + (i - c) \tag{3}$$

where *a* = arrowheads and *t* = tails occurring since the beginning of the structure, while *c* = commas and *i* = inverted commas occurring only on the tone.

**AM labels, mental representation, and structure.** The following demonstrations of IR will be presented alongside ToBI annotations to allow comparison with AM. The intonational features identified in AM have to do with information structure (for example, signaling whether the word adds new information or whether it refers to information that the listener should already know) and discourse function (for example, signaling whether a phrase ends a statement, prepares another phrase, or asks a question to which a response is expected). One may think of a discourse as an exchange in which the speaker helps the listener build and adjust a mental representation. Different types of accents signal how the listener should modify this representation. In many languages, including English, **H\*** typically marks information to be added as new, while **L\*** marks information that the speaker assumes the listener is already considering.

The autosegmental-metrical accents are aligned to a hierarchy of spoken tones (usually syllables) which receive more prominent stress, while phrase accents and boundary tones mark out a hierarchy of phrases. Boundary tones (**H%** or **L%**) mark the ends of intonational phrases (**IP**). Smaller segments called intermediate phrases (**ip**) are marked by phrase accents (**H-** or **L-**). In analyses of English, phrase accents often coincide with boundary tones, giving rise to notations such as **H-H%** and **L-L%** or, in shortened form, **H-%** and **L-%**. Such edge tones signal aspects of continuity between phrases. An **L%** boundary tone may convey that an IP is complete, while **H%** invites the listener to keep listening or to respond to a yes/no question. The tones labeled as pitch accents, boundary tones, and phrase accents in ToBI are hierarchically important; the rest are not labeled.

## Analyses

### Case 1: Phrase analysis. Four productions of *Marianna made the marmalade*

By comparing different ways of saying a phrase, we can learn more about the IR approach. Figs 7–10 show four different productions of the phrase *Marianna made the marmalade*. To introduce each production, the analyses include ToBI parsings for comparison. In Fig 7, *marmalade* (already present in the discourse) goes unlabeled, but *Marianna* (new information) receives **H\***. In Fig 8, **H\*** accents on both *Marianna* and *marmalade* mark each as new information. In Fig 9, a more abrupt rise labeled **L+H\*** marks *Marianna* as contrastive information that contradicts or revises the listener's representation. In Fig 10, the rising pitch at the end, **H-H%**, indicates a yes/no question.

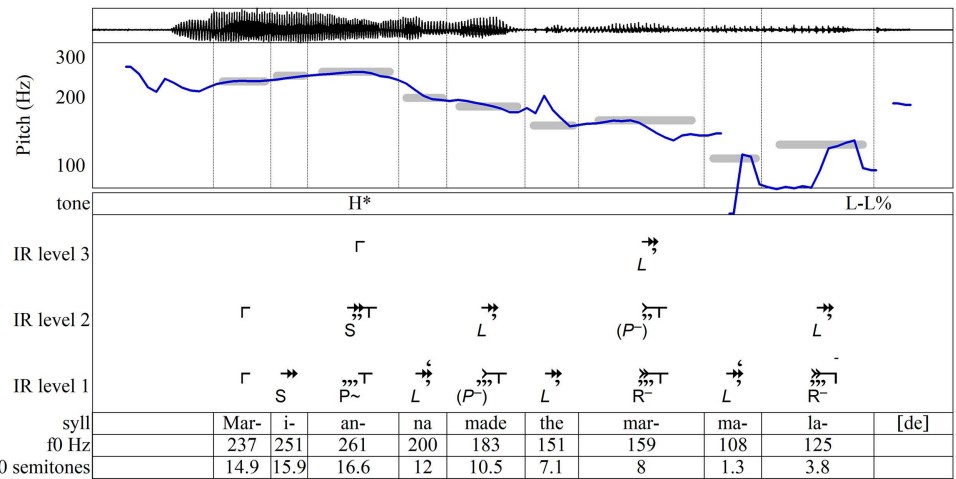

**Fig 7. "Marianna made the marmalade," uttered with *Marianna* as new information.** Recording and ToBI annotation from [19,20]. In all figures, semitones are in reference to 100 Hz.

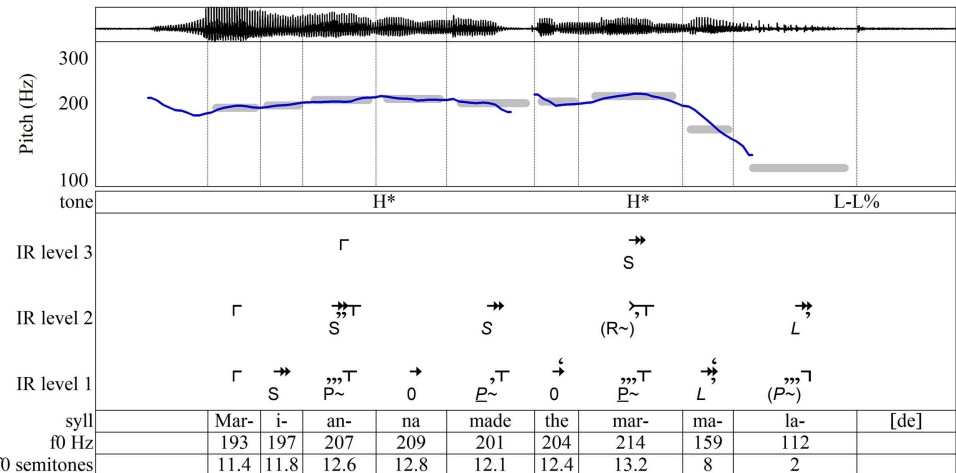

**Fig 8. "Marianna made the marmalade," uttered with *Marianna* and *marmalade* as new information.** Recording and ToBI annotation from [19,20].

We now turn to the IR parsings given in Figs 7–10. On the surface levels of all four of these examples, the closural tones are all substantially longer than the tones that precede them—often long enough to receive not just one but two or even three commas. Thus, it is the increased durations of these tones, rather than **R** or **–** intervals, that determine closure on Level 1's stressed syllables, *Mari<u>an</u>na <u>made</u> the <u>marmalade</u>*. By the nature of IR, however, deeper-level durations tend to vary less than surface-level durations, so **R** and **–** intervals more often contribute to the closing of groupings at deeper levels. For instance, in Fig 8, because the durational increase of *mar-* on Level 2 is only enough to generate one comma, the closural tail produced by **R** is necessary to end the grouping.

The tones that emerge deep in the IR hierarchies (i.e., in Level 3) occur on syllables carrying AM/ToBI pitch accents while the syllables that carry AM boundary tones and phrase accents appear only in the shallower IR levels (1 and 2). The stressed syllables of *Marianna* and *marmalade* emerge on Level 3 of each of the four different intonations of this phrase in

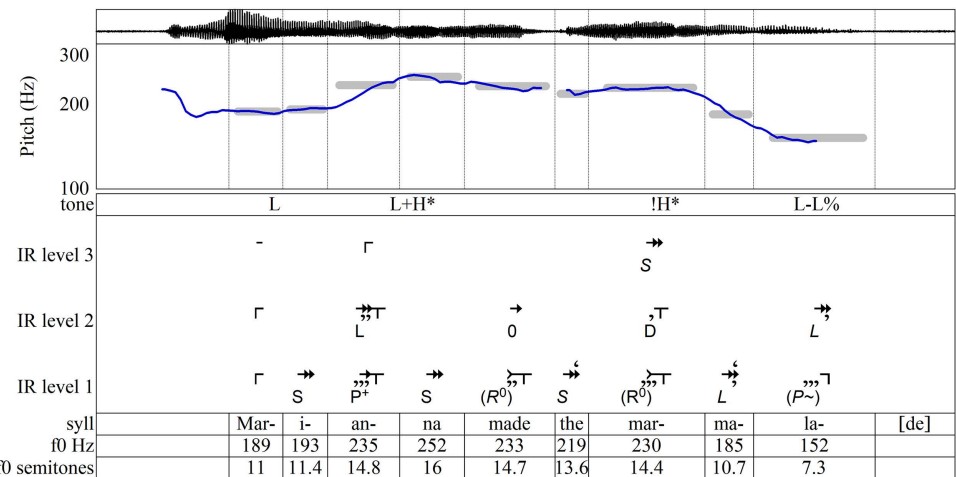

**Fig 9. "Marianna made the marmalade," uttered with *Marmalade* as surprising or contrastive information.** Recording and ToBI annotation from [19].

| tone | L | | L+H* | | | | !H* | | L-L% | |
|---|---|---|---|---|---|---|---|---|---|---|
| IR level 3 | - | | ⌐ | | | | →→ S | | | |
| IR level 2 | ⌐ | | L | | → 0 | | D | | L | |
| IR level 1 | ⌐ | S | L | S | (R⁰) S | S | (R⁰) | L | (P~) | |
| syll | Mar- | i- | an- | na | made | the | mar- | ma- | la- | [de] |
| f0 Hz | 189 | 193 | 235 | 252 | 233 | 219 | 230 | 185 | 152 | |
| f0 semitones | 11 | 11.4 | 14.8 | 16 | 14.7 | 13.6 | 14.4 | 10.7 | 7.3 | |

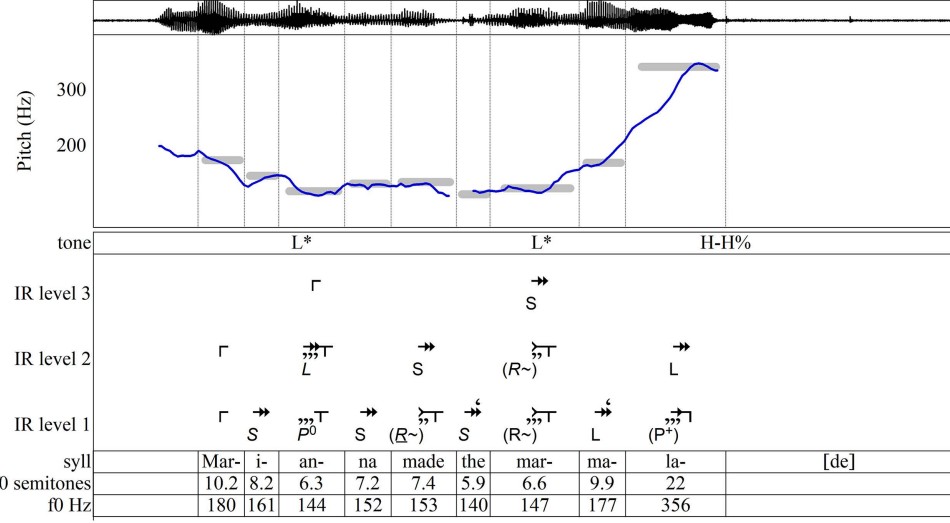

**Fig 10. "Marianna made the marmalade?" uttered as a yes-no question.** Recording and ToBI annotation from [19,20].

| tone | L* | | | | L* | | | | H-H% | |
|---|---|---|---|---|---|---|---|---|---|---|
| IR level 3 | ⌐ | | | | →→ S | | | | | |
| IR level 2 | ⌐ | | L | | S | (R~) | | | L | |
| IR level 1 | ⌐ | S | P⁰ | S | (R~) S | (R~) | L | (P⁺) | | |
| syll | Mar- | i- | an- | na | made | the | mar- | ma- | la- | [de] |
| f0 semitones | 10.2 | 8.2 | 6.3 | 7.2 | 7.4 | 5.9 | 6.6 | 9.9 | 22 | |
| f0 Hz | 180 | 161 | 144 | 152 | 153 | 140 | 147 | 177 | 356 | |

Figs 7–10, where they form implicative initiating intervals that vary from one production of the sentence to another. They may be interpreted as signaling the expectation of a tone to come in a later phrase. *Made* is also implicative, but only on Level 2, where it simply links the sentence's two most salient words. A different parsing would emerge if the sentence were intoned to confirm, say, that *Marianna made the marmalade* rather than buying it. This systematic correspondence between AM/ToBI and IR parsings may be interpreted as support for IR's approach. Beyond that, the IR parsing finds hierarchical salience on *made* and *marmalade* in every production, as well as placing them in relationship, with *made* subsidiary to the other two words.

When structural closure is achieved by increased duration, the intervals' properties can serve other functions. The **P+** on *Marianna* on Level 1 of Fig 9 may have a purpose other than keeping the grouping open, because secondary features

close it. Instead, it attaches expectancy to the meaning of the word *Marianna*. Thus **P+** corresponds to the **L+H\*** that, in AM's terms, marks new, revisionary information. Similarly, the **(P+)** that concludes Level 1 of Fig 10 acts as a highly implicative continuation interval (corresponding to AM's **H-H%** label). Even though the tone is closed, the expectancy expressed by **(P+)** adds a sense of uncertainty to *marmalade*, suggesting that the marmalade's status is central to this question. Conversely, in Fig 7, since commas denoting secondary effects are enough to bring about Level 1 closure on *mar-* and *-lade* as well as Level 2 closure on *mar-*, the **R–** intervals on these tones may be understood not merely as contributing to the closure of groupings but as attaching a meaning of finality to two syllables of the word *marmalade*— perhaps to close off further consideration of this mental representation. By comparison, as discussed above, ToBI leaves *mar-* unlabeled. Thus IR reveals an intricate interplay between two different functions of implication and closure. They not only help structure a multi-level hierarchy but also carry attributes tied to the lexical meanings of the words on which they occur.

### Method of labeling breaks for top-down processing

The utterances of Figs 7–10 are simple enough to be parsed entirely with IR's bottom-up principles, but more complex utterances require top-down interpretation. Key concerns for top-down interpretation are pauses and interruptions, which may be considered under the heading of *breaks*.

Breaks are an essential part of the ToBI scheme, which uses numbers from **0** (most conjunct) to **4** (most disjunct) to describe breaks. The numbers represent hierarchical reach: a break index (BI) of **1** represents a typically slight articulation between words, while 4 signifies a break separating intonational phrases [13]. In addition to such disfluencies, ToBI can mark deliberative pauses, as when a speaker is searching for a word. Such a pause is identified by the symbol **p** (standing for *prolongation*)—for example, **2p** for a deliberative pause between words or **1p** for a cut-off when a speaker stops to restart a word [18–20].

In IR, it is not viable to adopt AM's basic approach of assessing the magnitude or hierarchical reach of disfluencies or pauses. Listeners must use top-down judgement to distinguish, for instance, between breaks that leave implicative groupings incomplete, pauses that serve as "time-outs" while the speaker considers next words, and pauses that mark the ends of statements. The present adaptation of IR instead proposes several break symbols to capture the hierarchical reconfigurations brought about in the parsing of short silences, long pauses, broken-off phrases, interruptions, corrections, or restarts. These symbols represent top-down interpretations by the listener. Users of IRProsodyParser enter them into the top-down tiers of the analysis, and the computational algorithm interprets them as instructions specifying how treat silences and how to end or interrupt groupings.

Fig 11 details the types of breaks and their corresponding symbols. Some breaks have to do with silence. Interpreting a silence requires judging whether it occurs within or outside the temporal flow. For example, if a listener infers that a speaker is pausing to think, expectancy may continue—sometimes for many seconds—until the speaker continues. There are symbols for the two possible judgments. *Time-in* [**- - -**] adds the silence to the preceding tone's duration, producing the effect of a single long tone. *Time-out* [**…**] treats a pause as occupying no duration at all, effectively suspending time. When it removes the duration of a silence on its own level, it also shortens the corresponding tones on deeper levels. After a time-out or time-in, IRProsodyParser parses the following tone in relation to the previous tone, as though the silence were not present.

Other breaks enumerated in Fig 11 involve interruptions of normal implicative connections. In Narmour's terminology, one such interruption is *separation between groupings*, where, upon the close of a grouping, the start of the next grouping is delayed to the next tone instead of beginning on the closing tone. Both tones emerge on the next level. A second type is *suppression of implication*, where a grouping stops, unclosed, on an implicative tone without continuing. The next tone is heard as the start of a new grouping. Instead of a closing bracket, the final tone gets a square bracket (a grouping symbol also seen in Fig 5). The tones before and after the interruption are perceived not to be implicatively related to each other,

so no interval label is assigned. Finally, a broken square bracket indicates *releasable suppression of implication*, where the interruption is temporary and the tone following it is heard as continuing the grouping. The interval straddling the interruption is labeled as if there were no break.

Sometimes these implicative interruptions are signaled by silence or another feature, sometimes not. Either way, the type of interruption is usually a matter of top-down interpretation. In IRProsodyParser, the type is signaled by entering one of the symbols shown in Fig 11 in the top-down tier. A *separation break* [/] imposes separation and a *halt* [[]] imposes suppression. When a silence is involved, a *time-in* or *time-out* is usually given along with the break symbol, if only because this choice can affect the durational comparisons on higher levels.

| Imposed top-down interpretation | Symbol | Effect |
|---|---|---|
| Time-out | … | becomes |
| Time-in | --- or - | becomes |
| **Separation between groupings** | | |
| SEPARATION break (imposes closure followed by separation) | / | becomes / becomes |
| **Suppression of implication** | | |
| HALT break (imposes suppression followed by initiation) | | becomes |
| **Releasable suppression of implication** | | |
| SUSPEND or BOUNDARY break (imposes releasable suppression followed by close) | # or % | becomes # or % |
| Suppression of closure | & | |
| Imposition of closure | = | |
| Elevation of span to next hierarchical level | <> | |

⌐ tone

⅃ silence

**Fig 11. IR top-down break and elevation symbols.**

The *suspend* break is a bit more complicated. It imposes releasable suppression of implication on a tone, but it also directs the algorithm to impose a closing bracket on the following tone. This imposition arises because, when a time-out has interrupted the temporal framework, it may not be reasonable to ascertain secondary closure by comparing durations. Since *suspend* is often applicable at points where AM would use a boundary tone, the symbol [%] may also be used. When a silence is perceived as not affecting the structure of the utterance (for example, in a staccato delivery or a deliberative pause), a simple time-out break is sufficient.

Fig 11 lists three additional top-down symbols. It is often useful to use top-down *elevation* symbols [<] and [>]to move tones to the next hierarchical level. In case of need, IRProsodyParser provides two top-down symbols that simply suppress or impose closure on a tone.

**Case 2: implicative links between phrases. *Mother Teresa***

The parsings just seen in Figs 7–10 were generated using only the bottom-up rules of IR. Those of Fig 12 and Figs 13,14 illustrate important aspects of top-down interpretation and employ the symbols just introduced to parse levels, silences, and interruptions.

The question-with-answer analyzed in Fig 12(a) demonstrates how one phrase may connect to another on both shallow and deep levels. In the AM annotation, *example* ends with a **H%** boundary tone and *Teresa* with **L%**. In this IR parsing, the last syllable of each phrase is interrupted (*-ple* by a pause, *-sa* by the cessation of speaking) and left unclosed. These interruptions are *suspend* breaks [#]. The first releasably suppresses closure on *-ple*. The suppression is released when the speaker says *Mo-*, creating an important connection between the two phrases on the surface level as well as higher levels. (Here, Level 2 is the surface level. To ensure that each level contains tones of approximately similar durations while quickly spoken syllables remain on Level 1, longer tones are moved directly to Level 2 before parsing. The symbols <and> direct the computational algorithms to perform this adjustment. As a result, Level 2 sometimes represents the surface level.)

In a sign that IR's identification of closure is consistent with the linguistic intuition expressed in the ToBI annotation, the three **H\*** accents in the ToBI transcription seen in Fig 12(a) coincide with three of the four tones on Level 4 (*want, example, Teresa*). Two of these tones place focus on the words anchoring the most new information—*example* and *Teresa*. It is probably not coincidental that in IR terms these two words' stressed syllables complete intervals on Level 2 that thwart expectation. The stressed syllable of *example* completes an (**R˜**) interval in which the rising tone is unexpected because it follows a small interval. The stressed syllable of *Teresa* completes an **R+** interval, where **+** increases the expectancy rather than satisfying it, even while **R** and secondary features bring about closure. This **+** may be signaling that the attached word will revise the listener's mental representation. Such mirroring of information structure continues on deeper levels: a descending large interval **L** on Level 4 places *example* and *Teresa* in implicative relationship to each other, and the **L** is filled in on Level 3 by *Mother* to place an expectancy-increasing **P+** on *Teresa*.

At least one other interpretation is plausible enough for a listener to consider. The hearing parsed in Fig 12(b) interprets the pause after *example* as a *halt* instead of a *suspend*. This casts *Mother* on Level 2 and 3 as a restart, not as a release of suppressed implication. In this alternative hearing, the anchoring tones of the first phrase on Level 3 (*want* and *example*) form an interrupted implicative dyad so that the second phrase is heard as an echo in the form of another implicative dyad beginning on the same pitch (*Mother* and *Teresa*). In terms of phrasing, the interpretation of Fig 12(b) hears the question and answer as structurally parallel to each other, where that of Fig 12(a) hears *you want* as a lead-in followed by *example? Mother Teresa* as a three-element phrase. In terms of focus, this analysis treats *Mother Teresa* as salient new information, while the first emphasizes that *Mother Teresa* is an *example*. Notably, both interpretations place **P+** on the core syllable of *Teresa*—another case where a highly implicative continuation interval at a deep level anchors the utterance's most informative word.

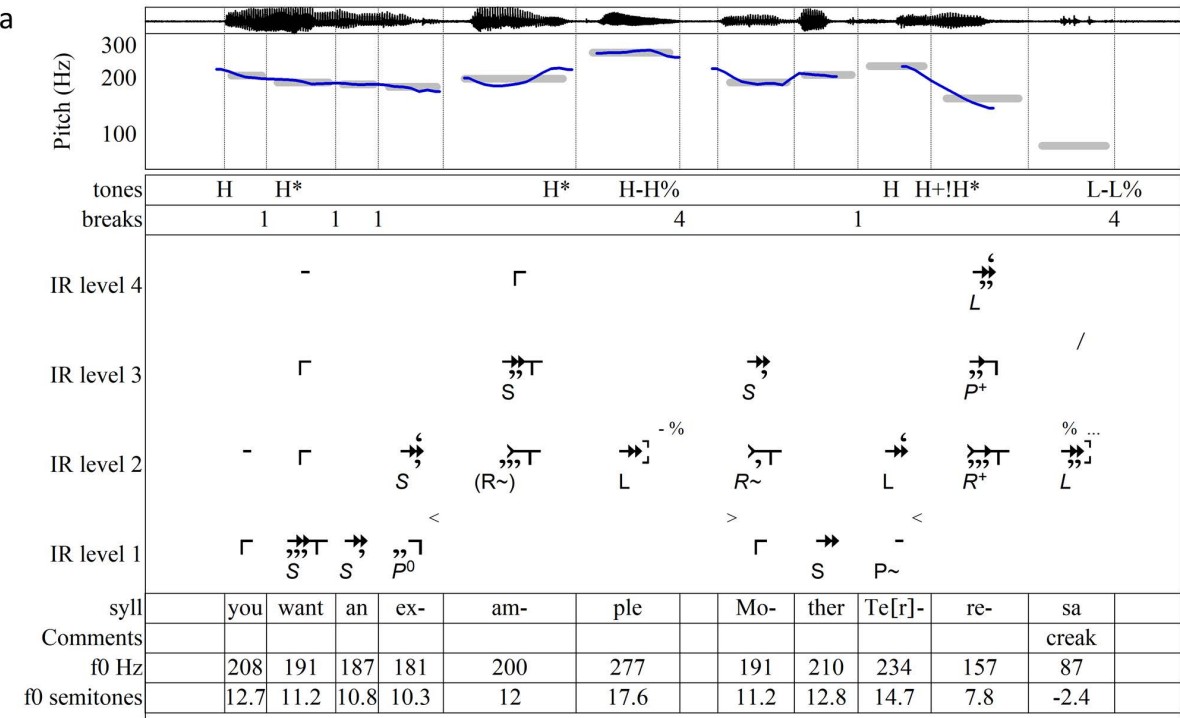

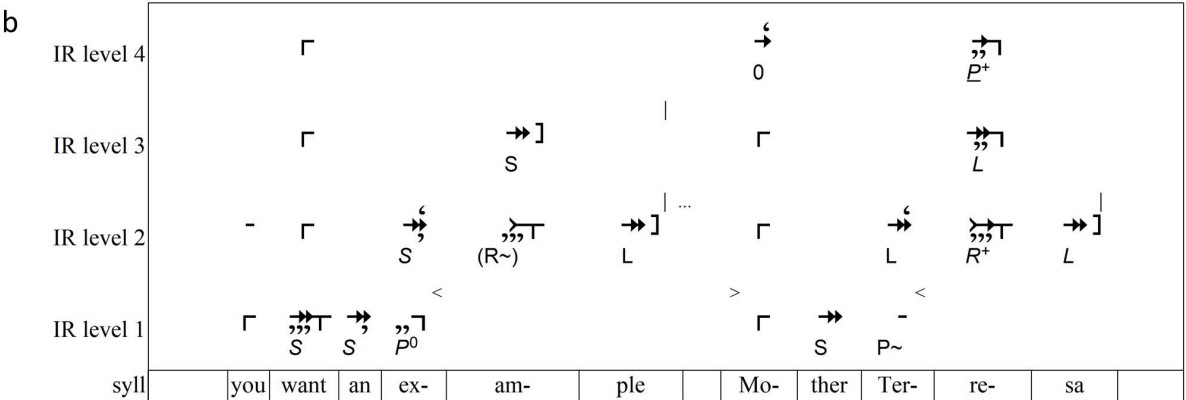

**Fig 12. Analyses of "You want an example? Mother Teresa" [20,70].** (a) ToBI annotation and IR parsing with *suspend* break after *example*. ToBI annotation from [20,70]. (b) IR parsing with *halt* break after *example*.

## Case 3: breaks in the parsing of *Pentagon*

Let us consider how different interpretations of breaks lead to different parsings of the utterance seen in Figs 13,14. For reference, the AM analysis of this utterance in Fig 13 finds **L+H\*** accents on *Pentagon* and *six*. The accent on *Pentagon* is probably used to indicate that the speaker is moving to a new topic; the accent on *six* is more puzzling. Following **L+H\*** on *six*, the**!H\*** downstep pitch accents on *southern* and *Iraqi* convey that *six southern Iraqi* is a series, in this case a list of modifiers applied to *cities*. The ToBI label **X\*?** indicates the annotator's uncertainty about what kind of pitch accent it instantiates.

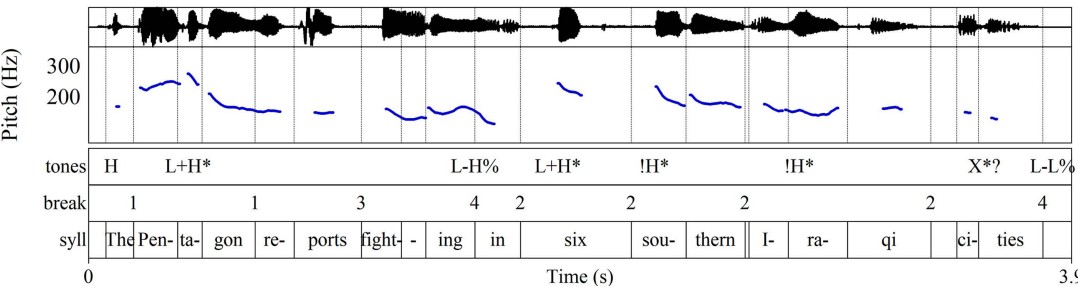

**Fig 13. ToBI annotation of "The Pentagon reports …" from [20,70].**

Fig 14(a) shows an IR parsing of the sentence without breaks (aside from time-in breaks [- - -] on the slight articulative silences after *southern* and *Iraqi*). This parsing has obvious flaws, including closure on weak syllables in *Pentagon*, *fighting*, *Iraqi*, and *cities*.

In keeping with the AM model, we may assume that stressed syllables should be closural so as to figure in deeper levels, and consecutive stressed syllables should be separated. Fig 14(b) achieves this by elevating the relatively slow tones of *in six* and *Iraqi* to Level 2 using **<and>**, thus populating each level with tones of similar duration. It also places *suspend* breaks [#] after the weak final syllables of *southern* and *cities* on Level 1 and *Pentagon*, *fighting*, and *Iraqi* on Level 2. Consecutive strong syllables in *reports fighting* and *six southern* are separated by separation breaks [/]. As a result of these adjustments, all points of closure on Level 2 are aligned with stressed syllables, so that each content word is represented on Level 3 by its strongest syllable: *The Pentagon reports fighting in six southern Iraqi cities.*

In the parsing of Fig 14(b), most ToBI breaks of 1 or 2 are represented as *suspends* [#], while the 3 becomes a separation [/]. However, with *Southern* extended into Levels 4 and 5 and separations before *Southern* on Levels 2 and 3, the parsing contradicts the ToBI annotation's BI of 4 between *fighting* and *in six*, and it goes against its **L+H*** marking *six* as more salient than *southern*. Fig 14(c) comes closer to the ToBI annotator's judgement through the addition of separations [/] between *fighting* and *in* on Levels 2 and 3. As a result, the deepest closural tone shifts to *six* from *southern* in Fig 14(c), and the syllables *six southern Iraqi cities* are grouped together on Level 3. While this parsing agrees with the placement of the AM pitch accent on *six*, it lacks the **P+** or **R+** interval that would support the label of **L+H*** (as opposed to a simple **H***).

IR avoids ToBI's uncertainty about the labeling of *cities*. Fig 14(b) and (c) both treat *cities* as **(R˜)** on Level 2 and **R–** on Level 3. Thus *cities* closes groupings on Levels 2 and 3 and appears on Level 4 in both parsings. The difference lies in whether, on Level 4, it completes an **S** begun with *southern* or an **L** begun by *six*. In this regard, Fig 14(b) may present a more plausible parsing than Fig 14(c). In Fig 14(b), the high pitch of *six* sets up a directional change that leads to a neatly closural **R–** on *southern* (Level 4). It registers the surprise on *six* associated with AM's **L+H*** label through an **(R+)** on Level 3.

The utterance examined here exhibits no significant audible break between *fighting* and *in*, nor does it present the retrospective parentheses or increases in implication that one would expect to correlate with the surprise of **L+H***. The ToBI break of 4 may reflect the intricate pitch motion on *fighting* that is symbolized **L-H%**, but it more likely reflects an assumption that the tone with the highest pitch, *six*, is the most salient. IR challenges this assumption. From the IR perspective, a pitch peak may simply prepare for the next tone to descend and create a closural **R**. This perspective may explain the pitch peaks on *six*, as it likely does in Fig 9, on the final, unaccented syllable of *Marianna*, which is assigned a ToBI **L+H***. Conventional AM would explain the latter's misalignment between the **H** pitch and the stressed syllable as a delayed peak

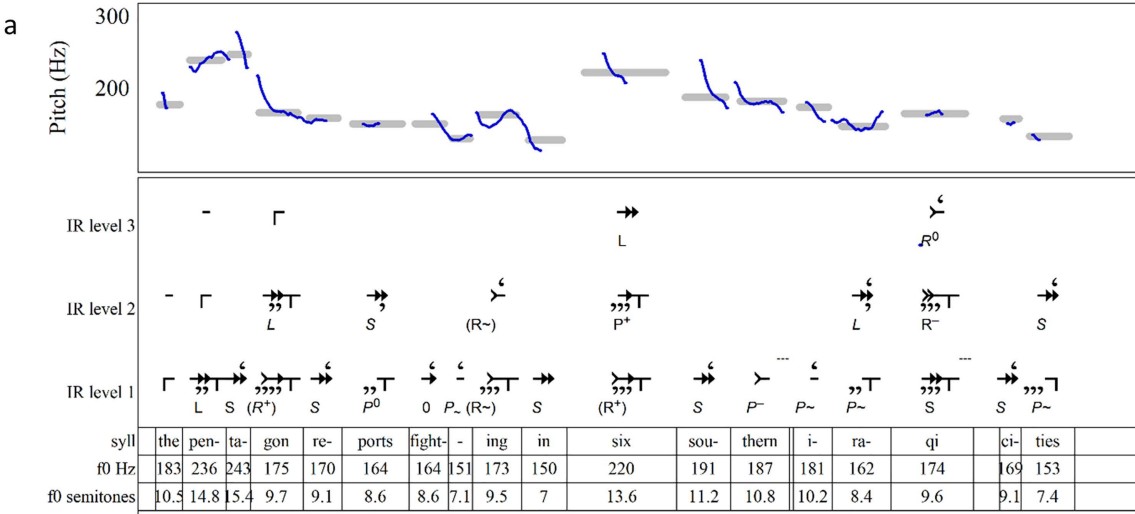
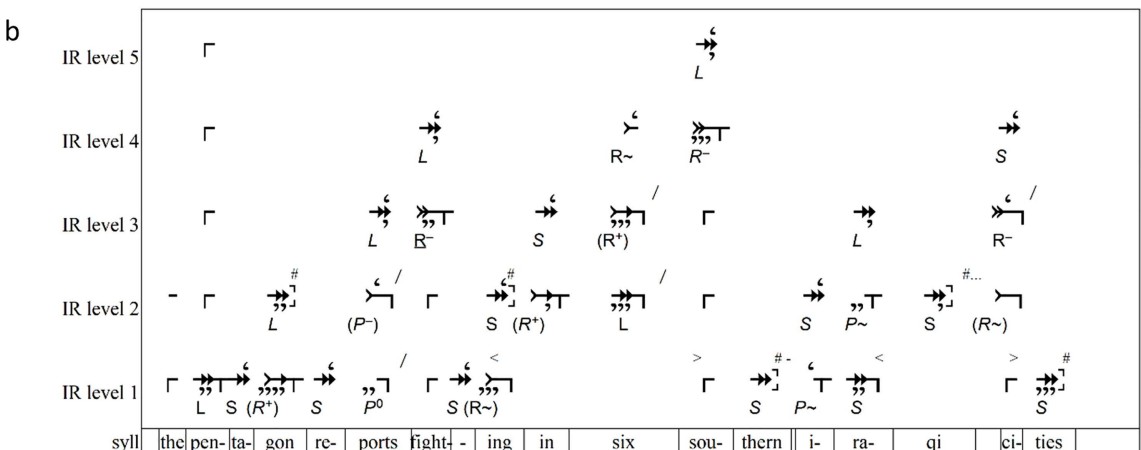
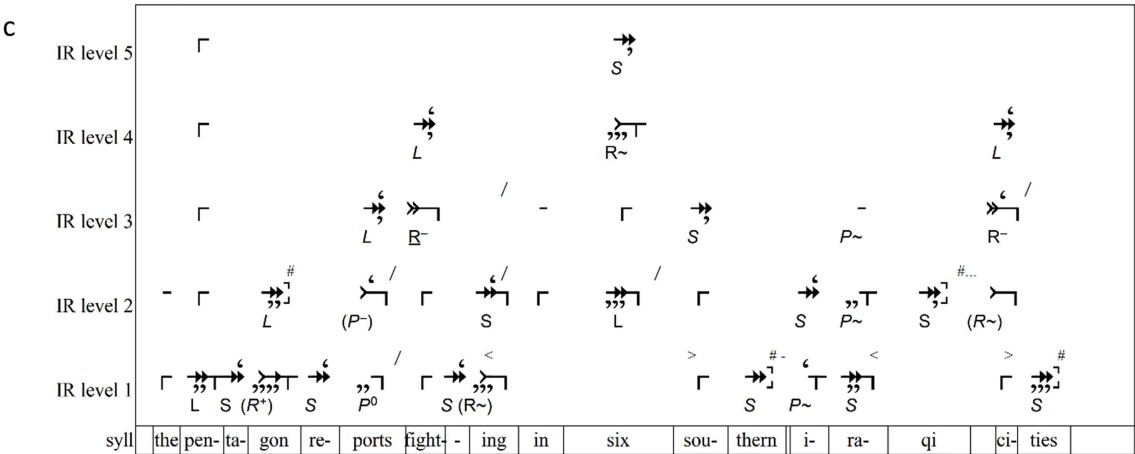

**Fig 14. IR parsings of "The Pentagon reports …"** (a) without breaks. (b) with top-down interpretations giving each content word a representation in Level 3. (c) with top-down interpretations placing *six* on the deepest level.

in which the speaker arrives late to the pitch peak that structurally belongs to the stressed syllable -*an*-. According to IR, though, the peak may be merely a pitch that creates the condition for the following (**R**) interval.

While other plausible parsings may be imagined, the two seen in Fig 14(b) and (c) demonstrate IR's systematic representation of the top-down interpretation of breaks. The two parsings have much in common. Notably, tones that AM regards as boundary tones—specifically *fighting* and *cities*—are positioned no deeper than Level 2, while the main semantic items of the phrase are placed on deeper levels. The uncertainty regarding the interpretation of breaks stems not from the speaker's intent or the structure of the utterance, but from whether the listener regards *six* to be more important than *southern*. Such top-down choices may well be guided by context as well as by individual and community linguistic practice.

## Discussion

IR offers a framework for understanding and describing melodic groupings, disjunctions, and contrastive distinctions based on the sizes and directions of intervals. Speech, as a form of harmony-free melody, provides an apt site for the application of IR's bottom-up approach to pitch. Building on this foundation, the foregoing analyses support the claim that IR's categories of interval comparison and closure may be productively applied to speech intonation. IR's parsings align with AM annotations: tones emerging at deeper levels of the IR hierarchy correspond to AM pitch accents, while syllables carrying AM boundary tones and phrase accents remain at shallower IR levels. This does not mean that IR is a proxy for AM. The IR parsings show many details not present in AM parsings, and IR proposes an explanatory framework very different from that of AM. In IR terms, tokens of the most informationally salient words project to deeper levels where they interact with each other, while shallower levels provide local connections that support the emergence of those deeper relationships, even when breaks interrupt the flow.

Certainly, this account of intonation and prosody requires further empirical study of IR. Rather than focusing narrowly on claims about pitch prediction, future experimentation should explore IR's proposals concerning closure and hierarchy. There is a need for experimental research addressing IR's account of hierarchical reduction, as well as the hypotheses advanced here regarding incomplete groupings, phrase connections, and the allocation of critical information to deeper hierarchical tones. As presented here, however, IR conforms to a number of accepted views about intonation. Specific tones within intonational contours align with words at critical junctures, contributing significantly to intonational meaning. Intonational contours operate on multiple hierarchical levels, not unlike the distinct intonational phrase and intermediate phrase of the AM model. Shallow-level closing and non-closing tones in IR's hierarchies map onto AM's pitch accents and boundary tones in predictable ways. Deep-level tones project backward and forward in a discourse, signaling when lexical items refer to concepts introduced earlier or project to new or contrastive concepts to be explored later.

At the same time, IR makes contributions in recognized areas of need. Not only does it provide a method for a fuller description and analysis of pitch intervals, duration, and potentially other parameters based on contrastible, distinctive perceptible categories, but its concepts of implication, expectancy, realization, and closure may form the basis of an explanation for the intonation contour's conveyance of information structure.

At its core, IR treats melodies as shaped streams of tones whose primary attribute is pitch-based expectancy, with pitch conceived in terms of a network of implicative or closural categorical interval relationships. The presence of such an implicative web in speech would explain how speech melody is able to carry out one of the principal functions of (intonation and) prosody: cohesion, which is the use of linguistic means to signal the relationships among ideas within a coherent discourse [12,16,71–73]. With melodic content thus understood in terms of implicative connections, intonational focus is seen to be conveyed through a mimetic correspondence—that is, one in which implicative connections are analogous to informational relationships. If it is borne out that implicative connections correspond to the information structure of a discourse, it may contribute to our understanding of the relationship between intonation and discourse; it has been suggested that IR has this potential [3]. Future research might inquire into a possible relationship between linguistic cohesion and IR's concept of implication.

While this article has focused on IR's categorical account of pitch and implicative structure, IR holds that listeners are always sensitive to interval comparisons whether or not they are large enough to be categorized as distinct **+** or **–** intervals. The model may thus address gradience in intonation, where variations in pitch within the same category can shift meaning. As an example of gradience, the connotation of AM's **L+H\* L-H%** intonation contour may shift from uncertainty to incredulity depending on the height of the **H\***. Such paralinguistic features often convey emotional information like surprise, doubt, or irony [4,8,24]. Ladd argues that gradience should be addressed through "fewer phonologically distinct intonational categories, and more dimensions of meaningfully gradient phonetic variation" [24]. Towards this end, IR operates by first comparing tones on a gradient before applying categorical boundaries—not to mention that its categorical boundaries are judgements which may shift depending on the context and the listener. **P+** or **(R+)** intervals, which often correspond to **L+H\*** accents, exemplify categories, but they are also effective because they are surprising and abrupt, and the degree of surprise varies with the size of the interval.

Given that prosody is multidimensional [24] and that pitch is not necessarily always one of its components [4], it is useful that IR does not limit its analytical purview to the domain of pitch. Narmour posits parametric scales for a variety of musical domains including duration, dynamics, timbre, and harmony [32], and even visual patterning [74]. For example, durational closure may be considered in terms of a comparative category called dynamic reversal (symbolized **dynR**). Similar principles may apply to intensity, timbre, and other features [32]. While parsing multiple implicative structures in parallel is conceivable, it is heuristically simpler—and perhaps cognitively more realistic—to treat one parameter, such as pitch, as primary, with other parameters evaluated as secondary.

IR analysis searches for hierarchical structure more than it catalogs sonic attributes. Its basic method involves evaluating tones' implicative openness or closedness. While these judgments draw on sonic features, they are shaped and mediated by top-down factors such as knowledge of a language or of what a speaker is trying to say. This may go against the expectation that intonation is governed by a transparent phonology. Each of the IR analyses in this study came together slowly and deliberatively. Perhaps this reflects the model's origins in the field of music theory and analysis. But even if future research offers insights that help narrow the analyst's choices, IR parsing may continue to seem much like critical reading. Indeed, it may be that a speech melody is a musical composition and should be interpreted as such. This is not a retreat to a position that says intonation is an art and is therefore mysterious, nor is it a claim that the slowness of analysis means that listeners cannot process intonation quickly. Intonational melody operates on the level of discourse as well as that of phonology. Uncertainty is to be expected in discourse, which involves speculations about others' mental representations.

However plausible IR may be as a hypothesis about how listeners perceive and react to the pitch and duration of each tone of every utterance, though, it seems unlikely that speakers deliberately compose each utterance tone by tone from scratch in real time. Linguistic Construction Grammar [75–77], an alternative to formal syntactic theory, holds that speakers build utterances from a large repertory of pre-formed, often non-compositional "constructions" whose functions and forms are learned together, rather than generating them solely from abstract, general rules. In music, too, composers use such constructions extensively [78–82]. By the same token, speakers likely rely on familiar, well-practiced intonational patterns, drawing on prior experience with known contours rather than calculating full implicative and hierarchical structures in the moment of production. IR, on its own, may therefore fall short in accounting for how intonation is actually produced. In spontaneous speech, speakers may aim for just a few salient pitch targets, filling in intermediate details through ingrained habits. Even if IR more successfully models those habits or captures the detailed conveyance of intonational meaning, AM-based analyses may more accurately reflect the realities of production.

In AM/ToBI analysis, determining when to label a high tone **H\***, **L+H\***, **H-**, or **H%** requires study of the utterance on its full scale and awareness of likely tone sequences. For instance, Hirschberg's schematic contour diagrams illustrate such likely sequences of tones [17,83]. In the sense that they systematize possible phrase-level AM configurations and describe the uses with which the more common configurations are paired, they may be seen as depicting constructions. Therefore we should not conclude that IR should supersede AM or other models; rather, this study advocates

pursuing them in parallel. Throughout this essay, the paried use of IR and AM has illustrated the benefits of this combined approach, with AM contributing top-down interpretations that inform IR's account of the syntagmatic and hierarchical relationships between tones.

Finally, this application of IR to speech should not be taken to show that language is musical. (It may just as well show the opposite: a way in which music is linguistic.) The preceding examples of speech intonation simply support the claim that IR's approach to interval comparison and closure may be productively applied to both. If, however, the account of intonation proposed in this article proves successful, and especially if IR proves useful to the study of human speech more broadly, it will illuminate the overlaps and non-overlaps between music and language. In this regard, much is to be gained from developing a single formalization that can be applied to both speech and music, such as the one proposed here.

## Supporting information

**S1 Text. A musical demonstration of the Implication-Realization model (IR).** Analyzing "Happy birthday to you.". (DOCX)

**S1 File. Pitch and time values used in the IR parsings.** Gives the pitch and timing values entered into the textgrids used by IR ProsodyParser in the preparation of each of the figures in the article. (CSV)

## Acknowledgments

The author thanks Ellie Bean Abrams, Sarah Wang, and Andrew Acs for testing earlier versions of IRProsodyParser and providing valuable suggestions; Melinda Yang for testing IRProsodyParser and thoroughly reviewing an earlier version of the manuscript; Aria Wang, YouYoung Kang, and two anonymous referees for careful readings; and students in his Spring 2025 Linguistic Elements of Music class at Pomona College who engaged seriously with the manuscript. The author also gratefully acknowledges Pomona College for a sabbatical subvention and for support of research assistants.

## Author contributions

**Conceptualization:** Alfred W. Cramer.

**Formal analysis:** Alfred W. Cramer.

**Funding acquisition:** Alfred W. Cramer.

**Investigation:** Alfred W. Cramer.

**Methodology:** Alfred W. Cramer.

**Project administration:** Alfred W. Cramer.

**Software:** Alfred W. Cramer.

**Visualization:** Alfred W. Cramer.

**Writing – original draft:** Alfred W. Cramer.

**Writing – review & editing:** Alfred W. Cramer.

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
