## [Decision Letter · Decision Letter 0]

23 Jun 2025

Dear Dr. Cramer,

Thank you for submitting your manuscript to PLOS ONE. After careful consideration, we feel that it has merit but does not fully meet PLOS ONE’s publication criteria as it currently stands. Therefore, we invite you to submit a revised version of the manuscript that addresses the points raised during the review process.

Two experts in the field have carefully reviewed the manuscript entitled “Intervallic intonation: The Implication-Realization Model of musical melody as representation of speech intonation and prosody“. You can find their comments below. I agree with their positive evaluations and consider your manuscript a valuable and thought-provoking theoretical contribution. However, reviewers suggest major revisions to

improve your manuscript before it can be considered for publication. Therefore, I invite

you to revise your manuscript and to respond to each review, although I will not request you to directly address the empirical validations required by one of the reviewers. Please pay attention to a thorough correction of typos and tables. Also, R2 notes that it is difficult to read through the analysis. Before analyzing full sentences, consider providing a "tutorial" section that walks the reader through a single, simple three- or four-note prosodic segment, explaining step-by-step how the symbols are assigned and how the hierarchy is built. 

Once again, thank you for submitting your manuscript to PLOS One and I look forward to receiving your revision.

We look forward to receiving your revised manuscript.

Kind regards,

Bruno Alejandro Mesz, Ph.D.

Academic Editor

PLOS ONE

Journal Requirements:

[The work was supported by a sabbatical subvention to AWC and support for a student research assistant, both provided by Pomona College (https://www.pomona.edu). No grant numbers were associated with this funding. The funder did not play any role in the study design, data collection and analysis, decision to publish, or preparation of the manuscript.].

3. In the online submission form, you indicated that [The data analyzed are available in publicly available sources cited in the article. Specifically, they are available in Brugos, A. et al (2006). Transcribing Prosodic Structure of Spoken Utterances with ToBI. MIT Open Courseware. https://ocw.mit.edu/courses/6-911-transcribing-prosodic-structure-of-spoken-utterances-with-tobi-january-iap-2006/pages/lecture-notes/. This source is cited at appropriate points in the article. The findings of the article do not depend specifically on such data, which are used for illustrative purposes.

In creating many of the illustrations, I have used computational scripts created by myself to be used in the linguistic software Praat. The scripts are available at acramer.sites.pomona.edu/IRProsodyParser. I am not including them as supplementary material here because I consider this article to be essentially a theoretical discussion of IR as applied to music and language. If it were deemed appropriate, however, I would be willing to consider including the scripts with the article.].

Reviewers' comments:

Reviewer's Responses to Questions

**Comments to the Author**

1. Is the manuscript technically sound, and do the data support the conclusions?

Reviewer #1: Yes

Reviewer #2: Partly

2. Has the statistical analysis been performed appropriately and rigorously?

Reviewer #1: N/A

Reviewer #2: N/A

3. Have the authors made all data underlying the findings in their manuscript fully available?

Reviewer #1: Yes

Reviewer #2: Yes

4. Is the manuscript presented in an intelligible fashion and written in standard English?

Reviewer #1: No

Reviewer #2: Yes

Reviewer #1: In a series of examples, the author shows how Narmour's IR system can be adapted to the analysis of speech intonation. It's an interesting paper. But I have a few major concerns that prevent my recommending publication in its current form:

1) As a proof-of-concept, the paper is convincing. But I thought it could be more direct and specific in identifying its implications. What sorts of specific problems in speech intonation identified in the literature could IR analysis potentially shed light on?

2) The Tables need to be revised:

a) they are incomplete. They don't contain all of the various symbols contained in the analyses (e.g., tilde [~] and number sign [#]). They must be carefully collated with the symbols in the figures and the body text.

b) the symbols in the tables are not consistent with those in the analysis. For example, the curly brackets shown in Table 1 don't appear in the analyses.

c) I assume the question marks (?) throughout Table 2 are a formatting error with the file, and not actual symbols. (See also, "Error! Reference source not found" in Table 2.)

All of these issues make it a chore to read through the analyses. For example, I really had to dig to know what R~ means on l. 559. I strongly recommend that the author consider the reader here. A "one-stop shopping" table where the reader can look up any unfamiliar symbol in the analyses would be helpful.

3) Although I did eventually read the whole thing, the author might benefit from knowing that, at some point on my first pass, I stopped reading the (very long) introduction and skipped to the data section. I anticipate many busy readers might do the same. For this reason, the author might consider moving some Introduction material to supplementary materials (e.g., ll. 193-274) and pointing the reader to those materials for further reading. Other solutions are possible (e.g., just trimming the Intro text).

Some other minor concerns:

1) l. 129-131 - Please provide supporting evidence for this empirical statement. Some contradicting evidence:

- Ammirante & Russo (2023), who showed that, whereas listener expectancy ratings on instrumental melodies were better predicted by relative pitch, ratings on sung melodies were better predicted by absolute pitch.

- Jacoby et al. (2019) showed that singers’ reproductions of sequences of tones tend to be distributed around their absolute pitch heights rather than transpositions that preserve their pitch relations

2) l. 184. Did Margulis really "introduce" the term "expectancy"? Not sure if Meyer used "expectancy" or "expectation". A search of google scholar shows the term "melodic expectancy" goes back at least as far back as Carlsen 1981. Or maybe I'm misinterpreting what you mean by "introduced"?

3) ll. 323-327. Not sure I follow the logic here (l. 325: "making it likely that [in speech] there are meaningful distinctions between intervals of very small sizes"). Aren't listeners constrained by JND for pitch? Aren't speakers constrained by intonation accuracy? E.g., studies from Pfordresher and Brown of singing without a pitch referent - a good analogue for pitch in speech - show that inaccurate and imprecise production of intervals are the norm. Also, if very small intervals are meaningfully distinct in speech, wouldn't we expect tone languages to have more tones than musical scales?

4) Related to 3), I found myself wanting to know the size of at least some of the speech intervals described in the analyses. The reason is that I'd like to be able to critically evaluate for myself claims of meaningfulness attributed to small speech intervals, which alternatively might just be meaningless variability in speech F0 production. This could be accomplished by having cents (or semitones) instead of Hz on the y-axis in the Praat screenshot, although I recognize that Praat might not do this natively.

5) for readers interested in potentially using IRProsodyParser, its description on pp. 431-436 could be more clear in terms of what it does and doesn't do. The author might consider describing it as semi-automated, and specify what needs to be done manually. I had to download and test it out to know.

6) ll. 485-488. Why not define the target pitch consistently? Also, please clarify whether it is the algorithm (automated) that is picking and choosing between average, local minima, etc., or the user manually making this decision.

Typos, etc.:

- ll. 57-58: "whether they signal whether" (awk)

- l. 72: "the H* and Marianna" - should this be "on" instead of "and"

- ll. 195-196: shouldn't this instead say "all tones appearing on Level 1 and nearly all on Level 1a"?

- l. 335: "arrows and tails" (needs a space)

- l. 392: "other features"

- l. 459: I don't know what "MAE_ToBI" is as compared to "ToBI" used elsewhere

- l. 523: isn't this describing Figure 10a rather than 10b?

- ll. 526-527: maybe so for the discrete pitch (long grey bars) but not for the raw contour (black line). For the raw contour, the pitch of "the" looks to be higher than for "mar".

- l. 543: "example" should be in italics

- l. 569: I don't see figure 11c anywhere and it isn't mentioned in the Fig. 11 captions (ll. 537-540)

- l. 743: I don't know what is meant by "mimetic" in this context

References

Ammirante, P., & Russo, F. A. (2023). Towards a Vocal Constraints Model of Melodic Expectancy: Evidence from Two Listening Experiments. Music & Science, 6.

Jacoby N., Undurraga E. A., McPherson M. J., Valdés J., Ossandón T., McDermott J. H. (2019). Universal and non-universal features of musical pitch perception revealed by singing. Current Biology, 29(19), 3229–3243.e12.

Reviewer #2: This paper introduces a novel methodological adaptation of the Implication-Realization (IR) model—originally conceived for melodic analysis—and applies it to the domain of linguistic prosody. The author offers an orthographically modified version of the model, tailored to speech, and demonstrates its use on English language examples via a custom Praat script (IRProsodyParser). The work is well-motivated in its desire to bridge music-theoretical and linguistic models of pitch structure. However, several issues limit the contribution in its current form.

Concerns:

1. Theoretical Alignment with AM Framework: One of the paper’s central claims is that IR-generated parsings align with prosodic elements identified in the Autosegmental-Metrical (AM) framework -- specifically, pitch accents, phrase accents, and boundary tones. While this alignment is intriguing, the paper does not explain why such agreement is meaningful or surprising. The AM framework is already a widely accepted and flexible system for representing intonation. If the IR model largely reproduces its structure, the question arises: what does the IR model add that AM does not already capture? Is the IR model merely a descriptive proxy, or does it offer theoretical or empirical advantages (e.g., what exactly can it predict)?

2. Limits of Musical Analogies in Speech Prosody: A more fundamental concern relates to the underlying assumption that principles derived from music perception (e.g., implication, realization, and transposability) can be directly ported to speech prosody. While both music and speech involve pitch, their functional and perceptual constraints differ. Musical melodies often assume transposability across registers or instruments, but speech prosody is anchored to a speaker’s modal pitch, with pitch variation serving different pragmatic and grammatical functions. The model does not appear to address how these vocal constraints may affect the viability of IR principles as applied to prosody.

3. Lack of Empirical Validation: Although the paper introduces the IR model as “cognitively based,” it does not engage with existing empirical literature supporting the psychological reality of hierarchical structures as described by IR. For example, in musical perception, Narmour’s IR model has received both support and critique in experimental studies. However, no such empirical link is drawn here for speech. The parsing examples (e.g., Figure 2) are intuitive, but without listener data or perceptual testing, it’s unclear whether the proposed hierarchical structures correspond to actual prosodic grouping by human listeners

4. Scope and Suitability for Journal Venue: While the paper may be a valuable contribution to methodological work in prosody, the format feels more like a theoretical exposition or tool introduction than an empirical study. For a journal like PLOS ONE, which emphasizes scientifically valid contributions, it would be helpful to include at least a minimal test of the IR model’s value. For example, a comparison with AM-based annotations on listener perception of prosodic boundaries, or a computational evaluation of model accuracy. In its current form, the work may be better suited to a journal focused on phonetics or theoretical linguistics.

**Do you want your identity to be public for this peer review?** For information about this choice, including consent withdrawal, please see our Privacy Policy

Reviewer #1: No

Reviewer #2: No

---

## [Author Response · Author response to Decision Letter 1]

22 Aug 2025

I believe all of my responses are contained in the revised cover letter and the reviewer response document that I uploaded. Please let me know if I should paste all of that here.

---

## [Decision Letter · Decision Letter 1]

10 Oct 2025

Dear Dr. Cramer,

Thank you for submitting your manuscript to PLOS ONE. After careful consideration, we feel that it has merit but does not fully meet PLOS ONE’s publication criteria as it currently stands. Therefore, we invite you to submit a revised version of the manuscript that addresses the points raised during the review process.

Dear Dr. Cramer,

Thank you for submitting your revised manuscript to PLOS ONE.

I have now received feedback from the reviewers, who are satisfied that the major concerns raised during the initial review have been thoroughly addressed. They were impressed with the extent of the revisions, and I am pleased to move forward.

Your manuscript is now close to being ready for acceptance. However, the reviewers have recommended a couple of final, minor revisions that I agree will enhance the manuscript's clarity and structure, making it more accessible to our broad readership.

Please submit the final revised version at your earliest convenience. We look forward to receiving it.

We look forward to receiving your revised manuscript.

Kind regards,

Bruno Alejandro Mesz, Ph.D.

Academic Editor

PLOS ONE

Journal Requirements:

Reviewers' comments:

Reviewer's Responses to Questions

**Comments to the Author**

Reviewer #1: (No Response)

2. Is the manuscript technically sound, and do the data support the conclusions?

Reviewer #1: Yes

3. Has the statistical analysis been performed appropriately and rigorously?

Reviewer #1: N/A

4. Have the authors made all data underlying the findings in their manuscript fully available?

Reviewer #1: Yes

5. Is the manuscript presented in an intelligible fashion and written in standard English?

Reviewer #1: Yes

Reviewer #1: Thank you for the opportunity to review the revision. The changes to the original ms appear to go far beyond addressing reviewer concerns. I have only focused on the concerns I raised initially, each of which has been adequately addressed in the revision.

I think there is still room to make the paper more amenable to a non-linear reader that wants to first quickly scan the paper for information related to their research interests before committing to reading from beginning to end:

- one or two "roadmap" sentences at the end of the last paragraph of the Introduction that specify what the method and case studies will accomplish

- where possible, the findings reviewed in the Discussion should be explicitly linked to a case study and/or figure (e.g., "(see Figure X)". For example, it looks like this could be done for the sentence beginning on line 766.

**Do you want your identity to be public for this peer review?** For information about this choice, including consent withdrawal, please see our Privacy Policy

Reviewer #1: No

---

## [Author Response · Author response to Decision Letter 2]

28 Oct 2025

There were two comments asking for minor revisions; I agree with both and have edited the manuscript accordingly. Please see uploaded response to reviewers and cover letter for details.

---

## [Editor Report · Decision Letter 2]

30 Oct 2025

Intervallic intonation: applying the Implication-Realization model of musical melody to speech intonation and prosody

PONE-D-25-17112R2

Dear Dr. Cramer,

We’re pleased to inform you that your manuscript has been judged scientifically suitable for publication and will be formally accepted for publication once it meets all outstanding technical requirements.

Kind regards,

Bruno Alejandro Mesz, Ph.D.

Academic Editor

PLOS ONE
---

## [Editor Report · Acceptance letter]

PONE-D-25-17112R2

PLOS ONE

Dear Dr. Cramer,

I'm pleased to inform you that your manuscript has been deemed suitable for publication in PLOS ONE. Congratulations! Your manuscript is now being handed over to our production team.

Kind regards,

on behalf of

Dr. Bruno Alejandro Mesz

Academic Editor

PLOS ONE